# RETRACTED: Protection of Mice from Controlled Cortical Impact Injury by Food Additive Glyceryl Tribenzoate

**DOI:** 10.3390/ijms24032083

**Published:** 2023-01-20

**Authors:** Suresh B. Rangasamy, Jit Poddar, Kalipada Pahan

**Affiliations:** 1Division of Research and Development, Jesse Brown Veterans Affairs Medical Center, Chicago, IL 60612, USA; sureshbabu_rangasamy@rush.edu; 2Department of Neurological Sciences, Rush University Medical Center, Chicago, IL 60612, USA; jit_poddar@rush.edu

**Keywords:** glyceryl tribenzoate, TBI, glial activation, lesion cavity, memory and learning

## Abstract

Despite intense investigations, no effective therapy is available to halt the pathogenesis of traumatic brain injury (TBI), a major health concern, which sometimes leads to long-term neurological disability, especially in war veterans and young adults. This study highlights the use of glyceryl tribenzoate (GTB), a flavoring ingredient, in ameliorating the disease process of controlled cortical impact (CCI)-induced TBI in mice. Oral administration of GTB decreased the activation of microglia and astrocytes to inhibit the expression of inducible nitric oxide synthase (iNOS) in hippocampus and cortex of TBI mice. Accordingly, GTB treatment protected and/or restored synaptic maturation in the hippocampus of TBI mice as revealed by the status of PSD-95, NR-2A and GluR1. Furthermore, oral GTB also reduced the size of lesion cavity in the brain of TBI mice. Finally, GTB treatment improved locomotor functions and protected spatial learning and memory in TBI mice. These results outline a novel neuroprotective property of GTB which may be beneficial in treatment of TBI.

## 1. Introduction

According to the Centers for Disease Control and Prevention, traumatic brain injury (TBI) is defined as a disruption in the normal brain function caused by a bump, blow or jolt to the head, or a penetrating head injury. In the United States, TBI is a leading cause of disability and death, particularly in children and young adults [1,2]. Each year, nearly 1.5 million people suffer from a TBI, with approximately 50,000 deaths from TBI related complications and around 85,000 people impacted by long term disabilities [3]. As stated by the Defense and Veterans Brain Injury Center, about 414,000 U.S. service members worldwide between 2000 and late 2019 suffered from TBI. Most of the TBI survivors suffer from different clinical symptoms, such as depression, cognitive/memory deficits, epilepsy and motor function impairments throughout the rest of their lives. Together, these leave a huge cost burden on society for the care and lost productivity due to TBI [4,5] and therefore, describing new and effective therapeutic approaches against TBI is an important area of research.

Activation of glial cells and synaptic damage play important roles in the pathogenesis of different neurodegenerative and neuroinflammatory diseases including TBI [6,7,8,9,10,11,12]. A number of proinflammatory molecules released from activated astrocytes and microglia are believed to cause synaptic injury [13]. For example, studies from focal and diffuse mouse models of TBI have revealed the involvement of various proinflammatory molecules such as TNF-α, IL-1β and inducible nitric oxide synthase (iNOS) in the disease process of TBI [11,14]. Accordingly, human studies showed the upregulation in IL-1β and TNF-α in Cerebrospinal fluid (CSF) and serum of TBI patients in comparison to healthy controls [15,16]. Recently we have described that glyceryl tribenzoate (GTB), a white crystalline chemical compound, has the potency to attenuate the expression of proinflammatory molecules and inhibit the pathogenesis of autoimmune and neurodegenerative disorders [17,18,19]. Oral GTB protects mice from experimental allergic encephalomyelitis (EAE), an animal model of multiple sclerosis (MS), via upregulation of TGFβ and regulatory T cells [18]. GTB also inhibits microglial activation in a mouse model of Huntington’s disease [17] and a monkey model of Parkinson’s disease [20]. GTB is used in the pharmaceutical industry as a safe and nontoxic additive in different dosage forms. Federal Emergency Management Agency (FEMA) has described GTB as a flavoring agent based on self-limiting properties, absorption, rapid metabolic detoxication and secretion in human and other animals.

Here, we examined the neuroprotective effect of GTB in a controlled cortical impact (CCI) mouse model of TBI. We demonstrate that after oral administration GTB is capable of attenuating glial activation, reducing the level of proinflammatory molecules, decreasing lesion volume, and improving synaptic structure in CCI-induced TBI mice. Functionally, oral GTB restored locomotor performance and improved learning and memory in TBI mice, highlighting possible therapeutic application of GTB in TBI.

## 2. Results

### 2.1. Attenuation of Astroglial and Microglial Activation in CCI-Induced TBI Mice by Oral GTB

Astrocytes and microglia are two important cell types of the central nervous system [6,7]. However, studies over the last three decades have revealed that upon activation, these cells release different proinflammatory molecules to participate in the pathogenesis of different neuroinflammatory and neurodegenerative disorders, including TBI [21,22,23,24,25,26,27].

Therefore, we examined the effect of oral GTB on glial activation in the CNS of TBI mice. At first, we monitored astroglial activation and as expected [11,12], CCI insult induced astroglial activation in cortex and hippocampus as revealed by enhanced GFAP expression on day seven post-injury as compared to sham control (Figure 1A,B). This finding was corroborated by counting of GFAP-positive cells in both cortex (Figure 1E) and hippocampus (Figure 1F). Increase in GFAP following TBI was further confirmed by Western blot analysis of hippocampal extracts (Figure 1I,J). Recently, we have seen that oral administration of GTB at a dose of 50 mg/kg body wt/d alleviates Huntington pathology in mice [17] and inhibits the adoptive transfer of EAE, an animal model of MS, in mice [18].

Therefore, here, CCI-insulted mice were treated with GTB orally via gavage at a dose of 50 mg/kg body wt/d and we observed a decrease in GFAP-positive astrocytes (Figure 1A–F) and the level of GFAP protein (Figure 1I,J) in the hippocampus of TBI mice upon GTB treatment. This result was specific as we did not find such change with vehicle treatment (Figure 1A–F,I,J).

Activated astrocytes express different proinflammatory molecules including inducible nitric oxide synthase (iNOS), which is known to produce excessive nitric oxide to cause nitrosative stress in a neuroinflammatory milieu [6,28]. Therefore, we examined the status of iNOS in the hippocampus and cortex of GTB-treated and untreated TBI mice. As expected, we also found increase in iNOS-positive cells (Figure 1A,B,G,H) and the level of iNOS protein (Figure 1K,L) in the brain of TBI mice as compared to sham control.

Many GFAP-positive astrocytes colocalized with iNOS (Figure 1A–D). However, similar to the suppression of astroglial activation, oral GTB also decreased iNOS-positive cells (Figure 1A,B,G,H) and the level of iNOS protein (Figure 1K,L) in the brain of TBI mice.

Next, we investigated microglial activation and found marked increase in Iba1-positive microglia in the cortex and hippocampus of TBI mice as compared to sham control (Figure 2A,B,E,F). This result was confirmed by Western blot of Iba1 in hippocampal extracts (Figure 2G,H). Double-labeling experiment also showed colocalization of Iba1-positive microglia with iNOS (Figure 2A–D). However, similar to the attenuation of astroglial activation, oral administration of GTB, but not vehicle, reduced the number of Iba1-positive astrocytes (Figure 2A–F) and the level of Iba1 protein (Figure 2G,H) in the brain of TBI mice. Together, these results suggest that oral GTB is capable of decreasing both astroglial and microglial activation in the hippocampus of TBI mice.

### 2.2. Oral Administration of GTB Reduces the Lesion Volume in the CCI Model of TBI

Since GTB treatment inhibited astroglial and microglial activation in the brain of TBI mice, we next decided to monitor whether oral GTB could reduce the lesion volume after 21 days post-injury. For measuring lesion volume, brain sections were stained with hematoxylin and eosin (H&E). Figure 3A displays H&E-stained brain sections arranged serially to show the volume of lesion cavity from different groups of mice. As anticipated, we found typical lesion with the distended cavity, originating from cortex through hippocampus and involving the lateral ventricle in TBI mice as compared to no lesion in sham control (Figure 3B). However, consistent with the suppression of astroglial and microglial inflammation, treatment with GTB, but not vehicle, reduced the size of lesion cavity in TBI mice (Figure 3A,B). This was also corroborated by quantitative analysis of lesion volume using the Cavalieri Stereological techniques, which revealed the decrease in total lesion volume in the whole hemisphere upon GTB treatment as compared to either untreated or vehicle-treated TBI-mice (Figure 3C).

### 2.3. GTB Treatment Restores Synapse Maturation in the Brain of CCI-Insulted Mice

Recent studies have shown that TBI has a major impact on synapse structure and function via a combination of the instant mechanical insult and the resultant secondary injury processes (e.g., inflammation), ultimately leading to synapse loss. For example, according to Witcher et al. [29], TBI causes chronic cortical inflammation mediated by activated microglia, ultimately leading to synaptic dysfunction.

Therefore, since GTB treatment reduces glial inflammation, we examined whether GTB could protect the synapse in TBI mice. PSD-95 is involved in synapse development and maturation [30]. Double-labeling of brain sections for NeuN and PSD-95 indicated loss of synaptic maturation in cortex and hippocampus of TBI mice as indicated by decrease in PSD-95 after 21 days post-injury in comparison to sham control mice (Figure 4A,B). On the other hand, we did not observe such loss of NeuN in cortex and hippocampus of TBI mice (Figure 4A,B). Western blot analysis of hippocampal tissues also confirmed a marked decrease in PSD-95 in the hippocampus of TBI mice as compared to sham mice (Figure 4E,F). However, consistent with the suppression of astroglial and microglial inflammation, treatment with GTB, but not vehicle, upregulated the level of PSD-95 in the brain of TBI mice (Figure 4A–F).

In addition to PSD-95, other molecules such as NR2A and GluR1 are also involved in synapse maturation [31]. Therefore, we also monitored the levels of NR2A and GluR1 and found significant decrease in both NR2A (Figure 4E,G) and GluR1 (Figure 4E,H) in the hippocampus of TBI mice after 21 days post-injury in comparison to sham control mice. Similar to the upregulation and/or restoration of PSD-95, GTB treatment increased the level of NR2A (Figure 4E,G) and GluR1 (Figure 4E,H) in the hippocampus of TBI mice. These results were specific as we did not observe any such increase in NR2A and GluR1 by vehicle treatment (Figure 4E,G,H). These results suggest that oral GTB is capable of restoring synapse maturation in the hippocampus of TBI mice.

### 2.4. Oral GTB Protects Cognitive Functions in TBI Mice

Many TBI survivors suffer from cognitive deficits throughout the rest of their lives [32,33]. It has been reported that impaired synaptic alterations are implicated in contributing to cognitive defects in TBI [34]. Since GTB treatment protected and/or improved synapse development and maturation in the hippocampus and cortex of TBI mice, we examined whether GTB could protect cognitive functions in TBI mice after 21 days post-injury. To monitor short-term memory, we employed novel object recognition (NOR), and for spatial learning and memory, mouse behaviors were analyzed on Barnes maze and T-maze.

As evident from the NOR task, TBI mice spent less time with novel objects as compared to sham control mice (Figure 5A,C). On the other hand, upon treatment with GTB, but not vehicle, TBI mice spent significantly more time with novel objects (Figure 5A,C), indicating improvement in short term memory by oral GTB. Barnes maze is a hippocampus-dependent memory task that requires spatial reference memory. It showed that TBI mice without treatments did not find the reward hole easily (Figure 5B), made more errors (Figure 5D) and required greater time (latency) (Figure 5E) as compared to sham control mice. However, GTB-treated, but not vehicle-treated, TBI mice performed much better on Barnes maze (Figure 5B), made fewer errors (Figure 5D), and took less time (Figure 5E) to find the target hole as compared to untreated TBI mice.

In T-maze as well, TBI mice without treatments exhibited fewer positive turns (Figure 5F) and more negative turns (Figure 5G) than sham control mice. Consistent with the NOR task and Barnes maze, oral administration of GTB, but not vehicle, considerably enhanced the hippocampus dependent memory performance in TBI mice as exhibited by a higher number of positive turns (Figure 5F) and a lower number of negative turns (Figure 5G) than untreated TBI mice.

### 2.5. GTB Treatment Improves Locomotor Functions in TBI Mice after Seven Days of CCI Injury

The principal therapeutic aim of TBI research is to preserve or recover behavioral function. Since GTB treatment protected cognitive functions in TBI mice, we next investigated whether GTB also protected overall locomotor activities. For recording general locomotor behaviors, we employed the Noldus computer system connected to a video camera 6 (Basler Gen I Cam—Basler acA 1300-60) that remained stationary on top facing-down on the open field arena. Figure 6A represents heat maps summarizing the overall movement of mice in the open field arena after seven days of CCI injury.

As expected, TBI mice exhibited decreased open field activity in comparison to sham control with respect to heat map (Figure 6A), distance travelled (Figure 6B), velocity (Figure 6C), center frequency (Figure 6D), and rearing (Figure 6E) on the seventh day post CCI injury. However, treatment of TBI mice with GTB, but not vehicle, led to significant increase in open field behavior (Figure 6A–E).

Next, we used rotorod test to examine motor coordination and balance activity of mice. Similar to open field activity, TBI mice exhibited significant decrease in latency to fall at seven days post CCI injury as compared to sham control (Figure 6F). On the other hand, oral administration of GTB, but not vehicle, improved rotorod performance as seen by increase in latency (Figure 6F).

Depression is a noticeable symptom of TBI particularly during the initial stage of brain injury, which can be monitored in mice by tail suspension test [35]. Therefore, we performed this test to monitor the effect of GTB treatment on depression-like behavior in TBI mice. As evident from Figure 6G, TBI mice on the seventh day of CCI insult exhibited significantly higher immobility time than sham control, indicating more depressive behavior in TBI mice than sham mice. However, GTB-treated TBI mice displayed significantly less immobility time during tail suspension test than either untreated or vehicle-treated TBI mice (Figure 6G), suggesting inhibition of depressive behavior by GTB.

TBI is known to damage the connection between brain and muscles, thereby impairing gait movements. Therefore, we employed beam walking to monitor gait behavior and observed poor gait movement of TBI mice as compared to sham control (Figure 6H–J). TBI mice used more steps (Figure 6H), took more time (Figure 6I) and made more slips (Figure 6J) than sham control mice while crossing the beam. However, oral administration of GTB, but not vehicle, improved beam walking of TBI mice (Figure 6H–J). To further confirm the results, we also used grid runway that allows scientists the opportunity to analyze and compare gait activities.

Similar to that found with beam walking, TBI mice also performed poorly in comparison to sham control on grid runway in terms of number of steps (Figure 6K), time taken (Figure 6L) and misplacement (Figure 6M). In this case as well, GTB treatment improved the performance of TBI mice on grid runway (Figure 6K–M). Together, these results indicate improved locomotor performance of TBI mice on the seventh day of CCI injury upon GTB treatment.

On the other hand, many of the locomotor parameters improved spontaneously on the 21st day of CCI injury and we also did not observe any significant change after GTB treatment (Figure 7A–M). For example, no significant change was seen in all parameters tested for open field behavior (Figure 7A, heat map; Figure 7B, distance traveled; Figure 7C, velocity; Figure 7D, center frequency; Figure 7E, rearing) and some parameters tested for beam walking (Figure 7H, number of steps; Figure 7I, time taken) and grid runway (Figure 7K, number of steps). Only on tail suspension test, was significant impairment seen in TBI mice as compared to untreated TBI mice and GTB treatment also led to significantly less immobility time during tail suspension test than either untreated or vehicle-treated TBI mice on the 21st day of CCI injury (Figure 7G), suggesting that GTB can inhibit depressive behavior even in the later phase of TBI.

## 3. Discussion

Apart from regular medical evaluation and care, effective treatment is still unavailable for improving the quality of life in patients with TBI. Therefore, delineating an effective, harmless therapy for modulating pathogenic processes and improving behavioral outcome is an important area of research. GTB is an FDA-permitted secondary food additive that is usually documented as safe for use in food or food packaging. Here we provide the first confirmation that oral gavage of GTB is capable of protecting mice from CCI-induced TBI. Although TBI caused a significant amount of lesion cavity, daily oral administration of GTB starting from 24 h after the CCI injury reduced the lesion volume and reestablished the structural-tissue integrity of damaged hippocampus. Similarly, GTB treatment also reduced motor dysfunction, improved cognitive performance and attenuated depression-like behavior in mice with TBI. Moreover, consistent with its safety profile, oral GTB did not cause any side effects (for example, bacterial or viral infection, decrease in body weight, loss of hair, fecal boli, unusual behavior, etc.). Together, these results suggest that oral GTB may exhibit beneficial effects for TBI and that GTB may not be toxic for patients with TBI.

Microglial and astroglial activation have become a hallmark of different neuroinflammatory and neurodegenerative disorders including TBI [11,12,36]. Usually following TBI, astrocytes and microglia are activated in and around the damaged area of the brain, which eventually produce different proinflammatory molecules (e.g., IL-1β, IL-6, TNFα, NO, etc.) to cause inflammation and synaptic injury [6,11,14]. Accordingly, although induction of TBI increased glial inflammation in the hippocampus as evident by upregulation of microglial marker Iba1 and astroglial marker GFAP and increase in the expression of inducible nitric oxide synthase (iNOS), oral GTB strongly inhibited and/or normalized astroglial and microglial activation. This is consistent with our recent findings demonstrating suppression of microglial inflammation in an animal model of Huntington disease [17] and hemiparkinsonian monkeys [20] by oral GTB. Several studies have shown that TBI has a major impact on synapse structure and function via a combination of the instant mechanical insult and glial inflammation, ultimately leading to synapse loss. For example, according to Witcher et al. [29], TBI causes chronic cortical inflammation mediated by activated microglia, ultimately leading to synaptic dysfunction. Consistent with the attenuation of glial activation, oral GTB restores synaptic maturation in cortex and hippocampus of TBI mice as indicated by increases in PSD-95, NR2A and GluR1. Therefore, by attenuating glial activation and restoring synaptic function, GTB can exhibit neuroprotection in TBI mice.

Mechanisms by which GTB can inhibit glial activation are poorly understood. Recently, we have seen that GTB is capable of suppressing the production of proinflammatory cytokines (TNFα, IL-1β and IL-6) in activated mouse primary glial cells [19]. Moreover, being a benzoic acid ester, GTB could be metabolized to benzoate in the body. For example, recently we have detected sodium benzoate (NaB), an FDA-approved drug against urea cycle disorders and nonketotic hyperglycinemia, in the cortex of GTB-fed Huntington mouse model [17] and in the substantia nigra of GTB-fed MPTP mouse model of PD [20]. NaB is known to inhibit activation of NF-κB and microglial inflammation via suppression of farnesylation–p21^Ras^ pathway [37]. Accordingly, oral GTB reduced the levels of GTP-bound p21^Ras^ (activated Ras) and p65 (RelA subunit of NF-κB) in the substantia nigra of hemiparkinsonian monkeys [20]. Therefore, in the hippocampus of TBI mice, GTB treatment may inhibit glial activation via modulation of the NaB–p21^Ras^–NF-κB pathway.

Despite intense investigations, there is no effective treatment for discontinuing the progression of TBI. Although some medications are available for blood clots, muscle spasms, anxiety, depression and mood instability, many of these drugs exhibit symptomatic relief with a number of side effects. Alternatively, there are several advantages of GTB over available anti-TBI therapies. First, GTB is a U.S. Food and Drug Administration (FDA)-permitted flavoring constituent that is used in food and food packaging businesses. Second, oral administration is the least painful route of drug treatment and GTB can be taken orally. Consistent with that found in mouse models of Huntington disease [17] and multiple sclerosis [18] and a monkey model of Parkinson’s disease [20], here, we have noticed reduction in glial activation in the hippocampus and protection of memory and learning in TBI mice by oral GTB. Third, many drugs do not cross the blood-brain barrier (BBB) to reach the brain. However, after oral GTB treatment, the active molecule, NaB, is detected in the brain [17], indicating its accessibility to the brain. Moreover, many TBI patients suffer from constipation and fecal incontinence [38]. Since GTB generates glycerol upon metabolism, it has a laxative effect that helps digested food move through the gut smoothly, therefore TBI patients may receive this as an additional benefit from oral GTB.

## 4. Materials and Methods

### 4.1. Animals

Male C57BL6 mice (7–8 weeks old) were purchased from Envigo, Indianapolis, IN for this study. Animal maintenance and surgical procedures were conducted in compliance with NIH guidelines for the Care and Use Committee and were approved by the Jesse Brown VA Medical Center Animal Care and Use Committee (protocol #1498771). Animals were housed in an environment with stable temperature and 12 h light-dark cycle. Water and food were provided ad libitum.

### 4.2. Controlled Cortical Impact (CCI) Injury

CCI injury was performed as described previously [27,39,40,41]. Briefly, mice were anesthetized with 2% isoflurane and allowed to breathe normally without tracheal intubation. During surgery, body temperature was maintained at 37 °C on a heating pad and monitored by a rectal probe. The head of anesthetized mice was shaved with sterile electric shaver and skin was cleaned with betadine solutions. A midline skin incision was performed to expose the skull and 4 mm diameter craniotomy was made in the right side of exposed skull with the coordinates −1.5 mm AP and −1.5 mm ML using the stereotaxic apparatus. Then the brain was exposed in this burr-hole with intact dura. Under surgical microscope control, the Leica Impact One Stereotaxic Impactor (Leica Microsystems, Buffalo Grove, IL, USA) attached with 1.0 mm rounded metal tip was angled vertically towards the brain surface with intact dura to cause mild CCI injury (1 mm tip and 1.25 V). Sham group animals underwent a similar surgical procedure but without CCI injury. The operated animal was then removed from the stereotaxic holder and the skin incision was lightly sutured to close the incised region. All operated animals were placed in a thermal blanket for the maintenance of body temperature. These animals were monitored until the recovery from anesthesia and over the next three consecutive postoperative days.

### 4.3. Treatment with GTB

GTB was solubilized in 0.1% methyl cellulose solution. Starting from 24 h of CCI injury, mice were treated with GTB (50 mg/kg in 100 µL volume) once daily via gavage for either 7 postoperative days or 21 postoperative days.

### 4.4. Experimental Groups

All mice were randomized into the following groups:

Group 1: Control/Sham: Mice underwent surgery without any injury and treatment.

Group 2: CCI: Mice underwent CCI injury and received no treatment.

Group 3: CCI + GTB: Mice with CCI received 100 µL GTB (50 mg/kg) daily via gavage.

Group 4: CCI + Vehicle: Mice with CCI received only 100 µL vehicle daily via gavage.

### 4.5. Determination of Group Size

Usually, any animal experiment is justified with a 99% confidence interval that generates *p* = 0.99 and (1 − *p*) = (1 – 0.99) = 0.01; ε is the margin of error = 0.05. Based on these values, the resultant sample size is as follows:
N=1.282∗0.991−0.990.052=1.282∗0.99∗0.010.052 =0.0160.0025=6.48~6


Therefore, six mice (n = 6) were used in each group.

### 4.6. Western Blotting

Western blotting was performed as described previously [21,25,26]. Equal amounts of protein samples were electrophoresed in 10% or 12% SDS-PAGE and transferred onto nitrocellulose membrane. The blot was probed with primary antibodies overnight at 4 °C. The following are the primary antibodies used in this study: anti-iNOS (1:1000, BD Biosciences, Franklin Lakes, NJ, USA), anti-Iba1 (1:1000, Abcam, Cambridge, UK), anti-GFAP (1:1000, Santa Cruz Biotechnology, Dallas, TX, USA), and anti-β-actin (1:5000, Abcam) (Table 1). Following the overnight incubation, primary antibodies were removed and the blots were washed with phosphate buffer saline containing 0.1% Tween-20 (PBST) and corresponding infrared fluorophore tagged secondary antibodies (1:10,000, Jackson Immuno-Research, West Grove, PA, USA) were added at room temperature. The blots were then incubated with secondary antibodies for 1 h. Following wash, blots were scanned with an Odyssey infrared scanner (Li-COR, Lincoln, NE, USA). ImageJ software (NIH, Bethesda, MD, USA) was employed for quantification of band intensities.

### 4.7. Immunohistochemistry

Mice were anesthetized with ketamine-xylazine and perfused with PBS and then with 4% paraformaldehyde (*w*/*v*) in PBS, followed by dissection of the brain [25,26,42]. Dissected brains were incubated in 10% sucrose for 3 h followed by 30% sucrose overnight at 4 °C. Brains were then embedded in optimal cutting temperature medium (Tissue Tech, Miami, FL, USA) at −80 °C and processed for conventional cryosectioning. Frozen sections (40 µm thickness) were treated with cold ethanol (−20 °C), washed with PBS, blocked with 2% BSA in PBST, and double labeled with two primary antibodies (Table 1). After three washes with PBST, sections were incubated with Cy2 and Cy5 (Jackson ImmunoResearch Laboratories). The sections were mounted and observed under an Olympus IX81 fluorescence microscope. Counting analysis was performed using Olympus Microsuite V software with the help of a touch counting module.

### 4.8. Quantification of Lesion Volume Using Stereological Techniques

The estimation of lesion volume was performed based on the Cavalieri method of unbiased stereology using the StereoInvestigator software (MicroBright Biosciences, Williston, VT, USA) [27,43]. Both the ipsilateral and contralateral hemisphere of brain volumes were determined using the Cavalieri estimator with a 1 mm grid spacing. Every fourth section was analyzed beginning from a random start point. Lesion volume was estimated by subtracting the volume of the ipsilateral hemisphere from that of the contralateral hemisphere. Then the volume of lesion cavity estimated in brain section of untreated mice was compared with lesion volume of brain sections of drug treated mice.

### 4.9. Behavioral Analysis

Analyses of behaviors in animals were conducted on the 7th and 21st postoperative days after CCI injury. These timepoints for behavioral testing were selected based upon earlier studies with these animal models where behavioral abnormalities were seen at these timepoints [27,44].

### 4.10. Open Field Behavior

The performance of animals in open field test was analyzed as described in our earlier studies [25,45,46]. Briefly, each animal was allowed to move freely to explore an open field arena designed with a square shaped wooden floor measuring 40 × 40 cm, with walls 30 cm high for 5 min. A video computer 6 (*Basler Gen I Cam—Basler acA 1300-60*) connected to a Noldus computer system was fixed in top facing-down on the open field arena. Each mouse was placed individually on center of the arena and the performance was monitored by the live video tracking system. The central area was arbitrarily defined as a square of 20 × 20 cm (half of the total area).

### 4.11. Rotarod

The fore–hindlimb motor coordination and balance in animals was observed using the rotarod test as described in earlier studies [45,47,48]. Briefly, each mouse was placed on the confined section of the rod and trial was initiated with a smooth increase in speed from 4 rpm to 40 rpm for 5 min. If the mouse did not fall from the rod, it was removed from the rod after 5 min. The latency to fall was measured in seconds and used for the analysis. Following the CCI injury, each mouse performed the task of three trials during the testing sessions and the average score on these three trials was used as the individual rotarod score. Each trial on the rod was terminated when the mice fell off the rod or held on to the rod by hanging and completed improper revolutions.

### 4.12. Tail Suspension Test

Mice were subjected to the tail suspension test as described previously [35,45,49]. The mice were gently hung upside down by the tail using non-toxic adhesive tape 50 cm above the floor for 6 min. Immobility time was defined as the period of time during which the mice only hung passively, without any active movements. An increased immobility time is defined as a depression-like behavior.

### 4.13. Nesting Behavior

This test was performed as described previously [27,45,50,51]. Briefly, a nestlet consisting of a 5 cm × 5 cm pressed cotton square was kept inside the cage between 5 pm and 6 pm. Next morning between 9 am and 10 am, two observers blind to our experimental procedures scored the quality of nest built by the mice using a 5-point scale as follows: Score 1 (>90% of nestlet intact), Score 2 (50% to 90% of nestlet intact), Score 3 (10% to 50% of nestlet intact but no recognizable nest site), Score 4 (<10% of nestlet intact, nest is recognizable but flat), Score 5 (<10% of the nestlet intact, nest is recognizable with walls higher than the mouse body).

### 4.14. Beam Runway

The beam runway is made of smooth wooden material and measures 65 cm length × 0.7 cm breadth × 4 cm height. A black box with an opening is fixed at one end and an aversive stimulus (bright lamp) at the other end of the beam. This test was used to evaluate the complex coordination and balance of mice while traversing the beam and we performed the procedure as described in earlier studies [45,52]. The mouse was placed on the beam near the light source and the light was turned on. This makes the animal move into the box to avoid the aversive stimulus, which was then turned off. Six repetitions were performed with a 2 min resting period inside the box. The parameters measured were the time taken (sec) to reach the box and the number of steps with contralateral limb drag/slips. An error was considered whenever the paw slipped on the beam and the number of slips were counted. The beam walk analysis was performed by an observer blinded to the treatment at 7th and 21st postoperative days.

### 4.15. Grid Runway

The grid runway (6 Fm length × 8 cm breadth × 1 cm intervals) made of parallel grid bars with interbar intervals of 1 cm apart and grid were kept above the surface of a table during the testing session [45,52]. A soft padding was positioned under the grid runway during the test for protection to avoid serious injury if the animal falls from the grid. Each mouse was allowed to walk freely on grid and the time taken and number of steps to cross the runway was noted. Each successful foot placement on the grid was recorded as a step. However, an error was considered whenever a paw sliped through the grid or the paw missed a bar and extended downward through the plane of bars. The locomotor behavior of the animal on the grid was evaluated by an observer blinded to the treatment.

### 4.16. Barnes Maze Test

The Barnes maze test was performed as described earlier [26,31,45]. Briefly, the mice were initially trained for two consecutive days followed by examination on day three. After each training session, maze and escape tunnel were thoroughly cleaned with a mild detergent to avoid instinctive odor avoidance due to mouse’s odor on the familiar object. On day three, a video camera (*Basler Gen I Cam—Basler acA 1300-60*) connected to a *Noldus* computer system was placed above the maze and was illuminated with high voltage light that generated enough light and heat to motivate animals to enter into the escape tunnel. The performance was monitored by the video tracking system (*Noldus System*). Cognitive behavior parameters were examined by measuring latency (duration before all four paws were on the floor of the escape box) and errors (incorrect response before all four paws were on the floor of the escape box).

### 4.17. T-Maze

The T-maze test was conducted as previously described [45,53]. Mice were initially habituated in the T-maze for two days under food-deprived conditions. Food reward was provided at least five times over a 10 min period of training. T-maze was cleaned with mild detergent solution between each testing session to minimize the animal’s ability to use any olfactory clues. The food-reward side was always associated with a visual cue. Each time the animal consumed food-reward was considered as a positive turn.

### 4.18. Novel Object Recognition (NOR) Test

This test evaluates the animal’s ability to recognize a novel object in the environment and monitor short-term memory as described in our earlier studies [45,53]. Initially, the mice were placed in a square novel box (20 in. long × 8 in. high) surrounded with an infrared sensor. Two plastic toys (2.5 to 3 in. size) that varied in color, shape, and texture were placed in specific locations in the environment 18 in. away from each other. The mice were able to freely explore the environment and objects for 15 min and were then placed back into their individual home cages. After 30 min intervals, the mice were placed back into the environment, with the two objects in the same locations, but now one of the familiar objects was replaced with a third novel object. The mice were again allowed to freely explore both objects for 15 min. The familiar and novel objects were thoroughly cleaned with a mild detergent after each testing session.

### 4.19. Statistical Analysis

Statistical analyses were performed with Student’s *t*-test for two-group comparison and one-way ANOVA or two-way ANOVA followed by Tukey’s post-hoc tests using GraphPad Prism 8. Data are represented as mean ± SD. Statistical significance was determined at the level of *p* < 0.05 [26,54].

## 5. Conclusions

We have described that oral administration of GTB, a flavoring ingredient and an indirect food additive, reduces microglial and astroglial activation, protects and/or restores synaptic maturation, and decreases lesion cavity to improve motor and cognitive functions in CCI-induced mouse model of TBI. Although the disease process of TBI in humans and mice are different, our results suggest that GTB may have therapeutic importance in TBI. Accordingly, GTB treatment protects and/or restores synaptic maturation in the hippocampus of TBI mice.

## Figures and Tables

**Figure 1 ijms-24-02083-f001:** Oral administration of GTB inhibits astroglial inflammation in vivo in the cortex and hippocampus of mice with TBI. TBI was induced in mice by CCI injury and after 24 h of injury mice were treated with 50 mg/kg/day of GTB via oral gavage. Seven days after GTB treatment, brain sections were double labeled for GFAP and iNOS ((**A**), control; (**B**), CCI; (**C**), CCI + GTB; (**D**), CCI + Vehicle). Cells positive for GFAP were counted in cortex (**E**) and CA1 region of hippocampus (**F**). Similarly, cells positive for iNOS were also counted in cortex (**G**) and CA1 region (**H**). Results represent analysis of six sections of each of six mice per group. Tissue extracts of hippocampal region from all groups of mice (n = 4 per group) were immunoblotted for GFAP (**I**) and iNOS (**K**). Actin was run as a loading control. Bands were scanned, and values (GFAP/Actin) (**J**) and (iNOS/Actin) (**L**) presented as relative to control.

**Figure 2 ijms-24-02083-f002:** Oral GTB decreases microglial activation in vivo in the cortex and hippocampus of mice with TBI. TBI was induced in mice by CCI injury and after 24 h of injury, mice were treated with 50 mg/kg/day of GTB via oral gavage. Seven days after GTB treatment, brain sections were double labeled for Iba1 and iNOS ((**A**), control; (**B**), CCI; (**C**), CCI + GTB; (**D**), CCI + Vehicle). Cells positive for Iba1 were counted in cortex (**E**) and CA1 region of hippocampus (**F**). Results represent analysis of two sections of each of six mice per group. Tissue extracts of hippocampal region from all groups of mice (n = 4 per group) were immunoblotted for Iba1 (**G**). Actin was run as a loading control. Bands were scanned, and values (Iba1/Actin) (**H**) presented as relative to control.

**Figure 3 ijms-24-02083-f003:** Decrease in lesion volume in TBI mice by GTB treatment. TBI was induced in mice by CCI injury and after 24 h of injury, mice were treated with 50 mg/kg/day of GTB via oral gavage. (**A**) Twenty-one days after injury, brain sections were stained with H&E and the stained sections were arranged in a series demonstrating the volume of lesion cavity in different groups. (**B**) Illustrative images of H&E stained sections are shown. (**C**) Lesion volume was quantified in all groups of mice. Statistical analyses were performed with two-way ANOVA and expressed as mean ± SD to compare the lesion volume between non-lesioned and lesioned side of the brain.

**Figure 4 ijms-24-02083-f004:** Restoration of PSD-95, NR2A and GluR1 in the hippocampus of TBI mice by oral administration of GTB. TBI was induced in mice by CCI injury and after 24 h of injury, mice were treated with 50 mg/kg/day of GTB via oral gavage. Twenty-one days after CCI injury, brain sections were double-labeled for NeuN and PSD-95 ((**A**), control; (**B**), CCI; (**C**), CCI + GTB; (**D**), CCI + Vehicle). Results represent analysis of one section of each of six mice per group. Hippocampal tissue extracts from all groups of mice (n = 4 per group) were immunoblotted for PSD-95, NR2A and GluR1 (**E**). Actin was run as a loading control. Bands were scanned, and values (Iba1/Actin, (**F**); NR2A/Actin, (**G**); GluR1/Actin, (**H**)) presented as relative to control. Data are expressed as mean ± SD. Statistical analyses were performed with one-way ANOVA.

**Figure 5 ijms-24-02083-f005:** Effect of GTB on spatial learning and memory in TBI mice. TBI was induced in mice by CCI injury and after 24 h of injury, mice were treated with 50 mg/kg/day of GTB via oral gavage. Twenty-one days after CCI injury, mice were tested by novel object recognition test ((**A**), Heat map; (**C**), Exploration time), Barnes maze ((**B**), Heat map; (**D**), number of errors; (**E**), latency or time taken) and T-maze ((**F**), positive turns; (**G**), Negative turns). Six mice were used in each group. Statistical analyses were performed by one-way ANOVA followed by Tukey’s post hoc test.

**Figure 6 ijms-24-02083-f006:** GTB treatment recovers motor functions in TBI mice. TBI was induced in mice by CCI injury and after 24 h of injury, mice were treated with 50 mg/kg/day of GTB via oral gavage. Seven days after CCI injury, mice were tested for open field behavior ((**A**), heat map analysis monitored by using the Noldus system; (**B**), distance moved; (**C**), velocity; (**D**), center frequency; (**E**), rearing), rotorod ((**F**), latency), tail suspension test ((**G**), immobility time), beam walking ((**H**), number of steps; (**I**), time taken; (**J**), slips), and grid runway ((**K**), number of steps; (**L**), time taken; (**M**), misplacements). Six mice were used in each group. Statistical analyses were performed by one-way ANOVA followed by Tukey’s post hoc test.

**Figure 7 ijms-24-02083-f007:** Effect of GTB on motor functions in TBI mice on the 21st day of CCI injury. TBI was induced in mice by CCI injury and after 24 h of injury, mice were treated with 50 mg/kg/day of GTB via oral gavage. Twenty-one days after CCI injury, mice were tested for open field behavior ((**A**), heat map analysis monitored by using the Noldus system; (**B**), distance moved; (**C**), velocity; (**D**), center frequency; (**E**), rearing), rotorod ((**F**), latency), tail suspension test ((**G**), immobility time), beam walking ((**H**), number of steps; (**I**), time taken; (**J**), slips), and grid runway ((**K**), number of steps; (**L**), time taken; (**M**), misplacements). Six mice were used in each group. Statistical analyses were performed by one-way ANOVA followed by Tukey’s post hoc test.

**Table 1 ijms-24-02083-t001:** Antibodies, sources and dilutions used in this study.

Antibody	Manufacturer	Catalog	Host	Application/Dilution
GFAP	Dako	Z0334	Rabbit	IF/1:2000
iNOS	BD Biosciences	610432	Mouse	IF/1:500
Iba1	Abcam	ab5076	Goat	IF/1:500
GFAP	Dako	Z0334	Rabbit	WB/1:1000
iNOS	BD Biosciences	610432	Mouse	WB/1:1000
Iba1	Abcam	ab5076	Goat	WB/1:1000
Actin	Abcam	ab1801	Mouse	WB/1:5000

IF, immunofluorescence; WB, western blot; GFAP, glial fibrillary acidic protein; iNOS, inducible nitric oxide synthase; Iba1, ionized calcium-binding adapter molecule 1.

## Data Availability

Not applicable.

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
