# Peer review of "Protection of Mice from Controlled Cortical Impact Injury by Food Additive Glyceryl Tribenzoate"

_ijms, 2023, doi:10.3390/ijms24032083_

Round 1

Reviewer 1 Report

Traumatic brain injuries (TBI) remain a major health hazard and a prominent risk factor not only for the development of post-traumatic stress disorder and dementia but also for neurodegenerative diseases especially Alzheimer’s disease (AD), Parkinson’s Disease (PD) and Parkinson’s Disease Dementia (PDD). TBI causes direct initial brain damage that primarily affects neuronal cells, astrocytes and blood vessels which then induce indirect secondary brain damage through neuroinflammatory responses.

Due to the current knowledge gap, a greater challenge is a lack of a simple, reliable, reproducible, and cost-effective therapeutic strategy to effectively treat TBI and decipher immediate and cumulative long-term molecular and cellular events post TBI treatment especially microglial activation, neuroinflammation, neurodegeneration, cognitive and motor impairment.

Research using animal models of TBI are extremely important to the development of novel drugs as well as repurpose existing FDA-approved substances. Repurposing already FDA-approved drugs or known non-toxic compounds that easily cross blood-brain-barrier (BBB) offer excellent opportunities for the rapid drug development for TBI.  

Authors of the current manuscript have identified glyceryl tribenzoate (GTB), a flavoring ingredient, that show anti-neuroinflammatory and neuroprotective properties in the well-established controlled cortical impact injury (CCI) mice model of TBI. This model mimics concussion, axonal injury, and BBB dysfunction and shows widespread neuropathological damage. This CCI model provides consistently reproducible data. 

Authors have clearly shown that oral GTB treatment protects and/or restored synaptic maturation in the hippocampus of TBI mice as revealed by the status of PSD-95, NR-2A, and GluR1. Furthermore, oral GTB also reduced the size of lesion cavity in the brain of TBI mice. They have also established that GTB treatment improved locomotor functions and improved cognitive performance and attenuated depression-like behavior in mice with TBI. These results demonstrated using several well-established experimental approaches, a novel neuroprotective property of GTB in which it may be therapeutic importance for TBI.  

The present studies provide the scientific rationale for the development of non-toxic therapy and has significant clinical implications and provides an efficient approach to test neuroinflammation-inhibitory compounds as therapeutic agents for TBI. Therefore, results presented are highly significant, innovative, novel and cost effective for both civilians as well as military TBI patients.

The manuscript adds timely to the mechanistic knowledge TBI. The work is original and has the potential for understanding the molecular biology of TBI. The experiments are well designed, the methods are rigorous, and the results are credible and impressive. The discussion is also appropriate. This paper represents a breakthrough in TBI research.

Author Response

Thank you very much for nice comments.

Reviewer 2 Report

This manuscript investigated the protection effects of a flavoring ingredient, glyceryl tribenzoate (GTB), in ameliorating the disease process of controlled cortical impact (CCI)-induced TBI in mice. The results of this manuscript suggesting GTB may be therapeutic importance for TBI. The manuscript has some important points and may be of potential interest to the reader. However, there are some flaws in the manuscript that affect the significance of the reported results.

1.     Figure 1 and Figure 2 are not very clear to read, mostly the brain sections.

2.     The title “Protection of mice from controlled cortical impact injury by food additive glyceryl tribenzoate” seems too huge, please revise to be more focus on the results the manuscript got.

3.     From the manuscript, we can see GTB decreased the activation of microglia and astrocytes to inhibit the expression of inducible nitric oxide synthase in hippocampus and cortex of TBI mice, and the results are supported by restored synaptic maturation by the status of PSD-95, NR-2A, and GluR1.After that, there are a lot of spatial learning and memory experiments of mice. I think the manuscript can go more deeper if the author can investigate more about the mechanisms of the relationship between the inflammation and the synaptic maturation.

4.     The last sentence of the introduction is too similar with the last sentence of the abstract, please revise either of them.

5.     The introduction kind of lack of the review of the biological properties of GTB, especially the effect of GTB on the neurological properties.

6.     If the effect of GTB on motor functions in TBI mice on 7 days pop and 21 days pop can be in the same figure, it will be more convenient for readers to compare.

7.     The amount of GTB is 50 mg/kg, actually as a food ingredient, usually in the daily use, we can’t use the huge amount, so the question is, why not choose smaller dose according to daily use amount of GTB?

8.     L671, the last sentence of “Conclusion”, is lack of a period.

Author Response

Comment: This manuscript investigated the protection effects of a flavoring ingredient, glyceryl tribenzoate (GTB), in ameliorating the disease process of controlled cortical impact (CCI)-induced TBI in mice. The results of this manuscript suggesting GTB may be therapeutic importance for TBI. The manuscript has some important points and may be of potential interest to the reader. However, there are some flaws in the manuscript that affect the significance of the reported results.

Response: Thank you for constructive comments. We have thoroughly addressed your concerns. For easy tracking, all changes are highlighted.

Other comments:

  1. Figure 1 and Figure 2 are not very clear to read, mostly the brain sections.
  2. The title “Protection of mice from controlled cortical impact injury by food additive glyceryl tribenzoate” seems too huge, please revise to be more focus on the results the manuscript got.
  3. From the manuscript, we can see GTB decreased the activation of microglia and astrocytes to inhibit the expression of inducible nitric oxide synthase in hippocampus and cortex of TBI mice, and the results are supported by restored synaptic maturation by the status of PSD-95, NR-2A, and GluR1.After that, there are a lot of spatial learning and memory experiments of mice. I think the manuscript can go more deeper if the author can investigate more about the mechanisms of the relationship between the inflammation and the synaptic maturation.
  4. The last sentence of the introduction is too similar with the last sentence of the abstract, please revise either of them.
  5. The introduction kind of lack of the review of the biological properties of GTB, especially the effect of GTB on the neurological properties.
  6. If the effect of GTB on motor functions in TBI mice on 7 days pop and 21 days pop can be in the same figure, it will be more convenient for readers to compare.
  7. The amount of GTB is 50 mg/kg, actually as a food ingredient, usually in the daily use, we can’t use the huge amount, so the question is, why not choose smaller dose according to daily use amount of GTB?
  8. L671, the last sentence of “Conclusion”, is lack of a period.

Responses:

1. Figures 1 and 2 have been enlarged. These two figures are clearly visible now.

2. Here, we have demonstrated beneficial effects of oral GTB on glial activation/inflammation, lesion volume, synaptic maturation, spatial learning and memory, and locomotor behaviors in TBI mice. The title we used is basically the summary of these findings in a few words.

If we mention these findings in title, it will look clumsy.

3. We have discussed the mechanisms. Please see 2nd and 3rd paragraphs of Discussion.

4. We have changed the last sentence of Abstract. 

5. We have done that. Please see lines 46 to 50.

6. We tried to combine these two. However, there are many parameters. These two will fit in one legible figure only if we discard some of the parameters.

7. For our EAE (animal model of multiple sclerosis) work (PMID: 28367355), we tested the effect of different doses of GTB on clinical symptoms of EAE and found optimum protection at a dose of 50 mg/kg body wt/d. Therefore, we have used this particular dose of GTB.

8. We have taken care of this. Thank you so much.